# Molecular Defense Response of *Bursaphelenchus xylophilus* to the Nematophagous Fungus *Arthrobotrys robusta*

**DOI:** 10.3390/cells12040543

**Published:** 2023-02-08

**Authors:** Xin Hao, Jie Chen, Yongxia Li, Xuefeng Liu, Yang Li, Bowen Wang, Jingxin Cao, Yaru Gu, Wei Ma, Ling Ma

**Affiliations:** 1School of Forestry, Northeast Forestry University, Harbin 150040, China; 2Key Laboratory of Forest Protection, National Forestry and Grassland Administration, Ecology and Nature Conservation Institute, Chinese Academy of Forestry, Beijing 100091, China; 3China Institute of Zoology, Chinese Academy of Sciences, Beijing 100101, China; 4School of Art and Archaeology, Zhejiang University, Hangzhou 310028, China; 5College of Pharmaceutical Sciences, Heilongjiang University of Chinese Medicine, Harbin 150040, China

**Keywords:** *Bursaphelenchus xylophilus*, pine wilt disease, nematophagous fungi, transcriptomic analysis, *Arthrobotrys robusta*

## Abstract

*Bursaphelenchus xylophilus* causes pine wilt disease, which poses a serious threat to forestry ecology around the world. Microorganisms are environmentally friendly alternatives to the use of chemical nematicides to control *B. xylophilus* in a sustainable way. In this study, we isolated a nematophagous fungus—*Arthrobotrys robusta*—from the xylem of diseased *Pinus massoniana.* The nematophagous activity of *A. robusta* against the PWNs was observed after just 6 h. We found that *B. xylophilus* entered the trap of *A. robusta* at 24 h, and the nervous system and immunological response of *B. xylophilus* were stimulated by metabolites that *A. robusta* produced. At 30 h of exposure to *A. robusta*, *B. xylophilus* exhibited significant constriction, and we were able to identify xenobiotics. *Bursaphelenchus xylophilus* activated xenobiotic metabolism, which expelled the xenobiotics from their bodies, by providing energy through lipid metabolism. When PWNs were exposed to *A. robusta* for 36 h, lysosomal and autophagy-related genes were activated, and the bodies of the nematodes underwent disintegration. Moreover, a gene co-expression pattern network was constructed by WGCNA and Cytoscape. The gene co-expression pattern network suggested that metabolic processes, developmental processes, detoxification, biological regulation, and signaling were influential when the *B. xylophilus* specimens were exposed to *A. robusta*. Additionally, bZIP transcription factors, ankyrin, ATPases, innexin, major facilitator, and cytochrome P450 played critical roles in the network. This study proposes a model in which mobility improved whenever *B. xylophilus* entered the traps of *A. robusta*. The model will provide a solid foundation with which to understand the molecular and evolutionary mechanisms underlying interactions between nematodes and nematophagous fungi. Taken together, these findings contribute in several ways to our understanding of *B. xylophilus* exposed to microorganisms and provide a basis for establishing an environmentally friendly prevention and control strategy.

## 1. Introduction

Plant-parasitic nematodes (PPNs) cause immeasurable losses to agriculture and forestry worldwide [1]. More than 4100 species of PPNs have been reported to be involved in restricting the safety and sustainable development of agriculture and forestry [2]. *Bursaphelenchus xylophilus* (pine wood nematodes, PWNs), a plant-parasitic nematode, causes pine wilt disease (PWD), which has caused serious ecological damage to forestry in East Asia and Europe [3]. In 2021, PWD was found in 731 counties of 19 provinces, with 14 million pines in 1.72 million hectares diseased and dead in China (os.bdpc.org.cn/, accessed on 1 January 2023) [4]. Synthetic nematicides are powerful tools used to manage PWD and are widely used in many Asian countries [5,6,7]. However, with the increased awareness of environmental protection and the requirements for sustainable development, there is a critical requirement for the development of sustainable methods to control PWNs.

Nematophagous fungi, a class of fungi that use nematodes as nutritional sources, have gained significant attention in the literature [8,9]. Nematophagous fungi are special microorganisms that use nematodes as a source of nutrition when the supply of their regular food source is limited. They are classified as nematode-trapping fungi, endoparasitic fungi, toxic fungi, and opportunistic fungi according to the way they infest nematodes [10]. Nematode-trapping fungi are the most widely studied, including *Arthrobotrys*, which forms contractile traps; *Monacrosporium*, which produces sticky traps; and *Dactylella*, which forms both contractile and sticky traps [11]. *Arthrobotrys*, one of the nematode-trapping fungi, has received extensive attention from researchers because the characteristics of this nematode’s interactions with *Arthrobotrys* are most easily observed. *Arthrobotrys* have been researched for a long time. Their preferred substrates, the formation and evolution of their traps, and their molecular mechanisms of pathogenicity against nematodes are only a few of the themes covered in the relevant studies [12,13]. Previous studies have shown that *Arthrobotrys* can be used to control plant-parasitic nematodes and animal-parasitic nematodes. For example, *A. cladodes* was used to control bovine-parasitic nematodes, regardless of the in vitro temperature changes [14]. *Arthrobotrys musiformis* protected sheep against gastrointestinal parasitic nematodes [15]. *Arthrobotrys oligospora* can be utilized to control the root-knot diseases caused by *Meloidogyne incognita* [16]. For the accurate and effective control of PPNs, researchers have concluded that elucidating the interactional mechanism between nematodes and nematophagous fungi is of considerable benefit [17]. In addition, the complete mitochondrial genomes of *A. oligospora* [18] and *A. musiformis* [19] have been described. These rich datasets have implied the conservation of the evolutionary relationship of *Arthrobotrys* across genomes [20]. *Arthrobotrys* activates multiple signal transduction pathways during the formation of traps for catching nematodes [21]. At present, the taxonomy, metabolites, and molecular functions of nematophagous fungi have been studied [22]. However, the molecular mechanisms of nematodes exposed to nematophagous fungi have not been fully characterized.

Technologies are constantly evolving, and omics studies constitute an essential tool for elucidating important mechanisms and metabolic pathways involved in the parasitism and pathogenicity of these fungi [23]. Gaps remain in our understanding of the complex trophic interactions between *Arthrobotrys* and PWNs. Transcriptomics, a high-throughput sequencing technology, can provide new insights into the *Arthrobotrys* control mechanisms of PWNs. In this study, PWNs were used to explore the molecular regulatory mechanisms of the nematode response to *A. robusta* by transcriptomic analysis. We investigated the physiological and molecular responses when PWNs were exposed to the nematophagous fungus. Based on gene expression, the method whereby the fungus affects the virulence mechanisms of PWNs was specifically demonstrated. This has important practical implications and extensional value for the biological control of PWNs.

## 2. Materials and Methods

### 2.1. Sample Preparation and RNA Sequencing

*Bursaphelenchus xylophilus* and nematophagous fungus were originally isolated from the xylem of diseased *Pinus massoniana* from Hangzhou, China. PWNs were cultured on *Botrytis cinerea* fungal mats at 25 °C for 7 days. Then, we collected PWNs and washed them with M9 buffer from the petri dishes [24]. The nematophagous fungus was characterized molecularly and morphologically and identified as *A. robusta*. Then, we sent it to the China General Microbiological Culture Collection Center (CGMCC) for preservation (CGMCC NO. 40258). The nematophagous fungus was cultured on low corn flour agar (LCMA) plates at 25 °C. Using 10 mL LCMA medium without a block as a control, 100 μL suspensions of PWNs (about 2000) were added dropwise to each plate. We recorded the nematophagous activity of *A. robusta* towards the PWNs every 6 h, and the number of rings of 200 PWNs was also recorded. Five replicates per time point were considered. Each replicate was counted 5 times. The phenotypic changes in the nematodes interacting with nematophagous fungus at 24 h, 30 h, and 36 h were observed using Zhang’s methods [25], and five replicates per time point were considered.

Total RNA was extracted with TRIzol (Invitrogen, ThermoFisher, Carlsbad, CA, USA). Subsequently, the quality of total RNA was checked by 1.1% agarose gel electrophoresis, and the purity was evaluated according to OD_260/280_ with the NanoPhotometer spectrophotometer 2000 (Thermo Fisher Scientific, Pittsburg, PA, USA), the Qubit 2.0 fluorometer (Invitrogen, USA) for accurate quantification of RNA concentration, and an Agilent 2100 Bioanalyzer (Agilent Technologies, California, USA) to accurately detect RNA integrity [26,27]. RNA-seq libraries were constructed with the NEBNext Ultra RNA Library Prep Kit for Illumina (NEB#7530, New England Biolabs, Ipswich, MA, USA) [28]. The mRNAs were purified from total RNA using oligo (dT)-attached magnetic beads [29]. Poly(A)-containing RNA was fragmented and used as the template for first-strand cDNA synthesis by the reverse transcriptase system M-MuLV (NEB# M0253L, New England Biolabs, Ipswich, MA, USA), followed by second-strand cDNA synthesis [30]. These cDNA fragments were then subjected to end repair; A-tailing was added, together with a ligation-sequencing adapter; and the fragments were then purified and size-selected by AMPure XP beads (Agencourt, Beckman Coulter, Brea, CA, USA) [31]. The cDNA of approximately 200 bp was screened for PCR amplification, and the PCR products were purified again with AMPure XP beads to obtain the final cDNA library [32]. High-throughput sequencing was performed using the Illumina Nova-seq 6000 developed by Gene Denovo Biotechnology Co. (Guangzhou, China) [33].

### 2.2. Analysis of Raw Data

To ensure the data quality of RNA-seq, the raw data were filtered before the information was analyzed to reduce analytical interference from invalid data. Raw reads for all sequenced libraries were quality controlled by HISAT2 to filter the low-quality data (>50% of the bases with a quality value Q ≤ 20) and thus obtain clean reads [34]. The clean reads were then mapped to the reference genome (https://parasite.wormbase.org/Bursaphelenchus_xylophilus_prjeb40022/Info/Index/, accessed on 17 August 2022) [35]. Quantitative calculations were performed using fragments per kilobase per million reads (FPKM) to obtain gene expression [36]. Significantly differentially expressed genes (DEGs) were defined as genes for which *p* < 0.05 and |log_2_FC| ≥ 2 according to the DESeq2 package [37].

Gene annotation was performed with the Kyoto Encyclopedia of Genes and Genomes (KEGG) database (http://www.genome.jp/kegg/, accessed on 30 August 2022), Gene Ontology (GO) annotation (http://www.geneontology.org/, accessed on 30 August 2022), and the NCBI nonredundant protein (Nr) database. GO and KEGG enrichment analyses were performed with the hypergeometric distribution algorithm in ClusterProfiler package (https://bioconductor.org/, accessed on 30 August 2022). RNA differential expression analysis was performed with DESeq2 (R package). Genes with a false discovery rate (FDR) ≤ 0.05 and absolute fold change ≥2 were considered differentially expressed genes.

### 2.3. Analysis of Differentially Expressed Gene (DEG)

Gene Set Enrichment Analysis (GSEA) can effectively compensate for the lack of effective information mining of micro-effective genes in traditional enrichment analysis and provide a more comprehensive explanation for the regulatory role of the GO term or KEGG pathway. GSEA was performed on all genes in the comparator group to obtain and interpret patterns of gene changes in several biologically important pathways. Significantly enriched pathways in the comparison group and the correlation of pathways with different subgroup samples can be identified based on certain ES and *p* value thresholds [38].

To identify whether genes in a predefined functional gene set exhibited significant or consistent differences between the two biological states, we performed gene set enrichment analysis using the KEGG database for all genes in a single comparison group via GSEA v4.2.3 software (Eric Lander, Cambridge, MA, USA, http://www.broadinstitute.org/gsea, accessed on 30 September 2022). The processes of GSEA were uploaded, with the expression files classified as the expression dataset [39], enriched genes selected as the gene set database, 1000 selected as the permutation number, phenotype files used as phenotype labels, no collapsing applied to facilitate the comparison of the order, and gene set size filters applied (min = 15, max = 500). The results were analyzed by screening enrichment pathways based on |NES (normalize enrichment score) | > 1, NOM *p* < 0.05, and FDR *q* < 0.05.

### 2.4. Construction of the Weighted Gene Co-Expression Network Analysis (WGCNA)

A gene co-expression network was constructed via weighted gene co-expression network analysis (WGCNA) in R [40]. Hierarchical clustering analysis was performed based on the weighted correlation. Finally, *β* = 7 was selected as the threshold for filtration. The adjacency matrix was then transformed into a topological overlap matrix (TOM) to evaluate the correlation between gene expression levels [35], and the dissimilar topological matrix (dissTOM = 1-TOM) was used to carry out matrix clustering and module partitioning via the dynamic sharing algorithm. The minimum number of elements in a module was 50 (module size = 50), and the threshold for the merging of a similar module was 0.3 (CutHeight = 0.3). The Pearson correlation coefficient of the corPvalueStudent () function was used to calculate the correlations between the binding matrix and the module feature gene matrix to obtain the *p* value. Binding-related specific modules were identified based on |r| > 0.60 and *p* value < 0.001 as specific modules for subsequent analysis. The networks were visualized using Cytoscape_3.9.1.

### 2.5. Validation of Gene Expression by qRT-PCR

The RNA was reverse-transcribed by Prime Script™ IV 1st strand cDNA Synthesis Mix (Takara Biomedical Technology, Beijing, China). The genes related to growth and development, xenobiotic metabolism, and pathogenicity were subjected to quantitative, real-time polymerase chain reaction (qRT-PCR) using 2 × SYBR Green qPCR Master Mix (Bimake, Shanghai, China) [41] (Appendix A). Three technical replicates and three biological replicates were tested for each sample and analyzed using the 2^−ΔΔCt^ method [42]. To compare the RNA-Seq and qPCR results, a linear correlation was calculated using Log_2_FC.

### 2.6. Data Analysis

Data were plotted and analyzed by Prism version 9.0 (GraphPad, San Diego, CA, USA). All data are expressed as the mean ± standard error (SE). Logarithmic or square root transformations were used to improve data normality or variance uniformity. Differences between group data were tested for significance by one-way analysis of variance (ANOVA) [43]. Statistically significant differences between the treatment and control are indicated by different letters (*p* <  0.05).

## 3. Results

### 3.1. A. robusta Identification and Interactions with B. xylophilus

*Arthrobotrys robusta* was isolated from PWN-infected *P. massoniana*. The mycelium was white and sparse. Conidiophores were solitary and unbranched, with the base contracted towards the end, the tips not expanded, and forming multiple dentate peduncles, each with two to seven conidia (Figure 1A–C). When *A. robusta* and PWNs were cocultured, the conidia fell off, and the hyphae formed three-dimensional cords to bind the nematodes (Figure 1D). The conidia were colorless, oval or ovate, with a broad and rounded upper part and a contracted, flat, and truncated lower part. The septum was not constricted and had septate spores [18.3–21.3 × 7.5–9.8 μm (n = 50)] and amerospores [13.2–18.6 × 5.7–8.8 μm (n = 50)] (Figure 1E–G). At 24 h, PWNs were found inside the traps of *A. robusta,* and the nematode-trapping efficiency was 6.47% (Figure 2A). At 30 h post-inoculation, the trap structures contracted and fixed the nematodes. After 36 h, the movement of the PWNs was restricted by several trap structures and the nematode-trapping efficiency was 24.42% (Figure 2B), with the color of the nematodes’ bodies being significantly darken, and the nematode-trapping efficiency was 29.03% (Figure 2C,D).

### 3.2. RNA Sequencing and Gene Expression Patterns

A total of 16,252 genes were predicted by mapping them to the reference genome of *B. xylophilus*. Among these, 15,884 (97.74%) genes were annotated in the Wormbase database, 7416 (45.63%) genes were annotated in the nonredundant protein database, 10,609 (65.28%) genes were annotated in the SwissProt database, 3248 genes (19.99%) were annotated in the KEGG database, and 11,599 (71.37%) genes were annotated in the GO database.

There was a high level of independence of the data among each sample (Appendix A and Appendix A). There were 722, 694, and 765 DEGs at 24 h, 30 h, and 36 h, respectively, compared with the control group (Figure 3A). A total of 505 shared DEGs between the three different time points with the control samples were analyzed (Figure 3A). Metabolic process (226 DEGs, GO:0008152), response to stimulus (113 DEGs, GO:0008150), and biological regulation (103 DEGs, GO:0065007) were classified as level 2 (Figure 3B). Structural constituent of the cuticle (48 DEGs, GO:0042302), acetaldehyde metabolic process (7 DEGs, GO:0006117), ethanol catabolic process (7 DEGs, GO:0006068), and collagen trimer (30 DEGs, GO:0005581) were significant GO terms according to GO enrichment analysis (Figure 3C). Moreover, there were 117 DEGs divided into 13 KEGG pathways by KEGG (*p* < 0.05, *q* < 0.05). Among them, xenobiotic biodegradation and metabolism contained major DEGs according to KEGG analysis, such as drug metabolism by cytochrome P450 (26 DEGs, ko00982), metabolism of xenobiotics by cytochrome P450 (25 DEGs, ko00980) (Figure 3D). Additionally, membrane transport of environmental information processing included three ABC transporter genes B, and signal transduction of environmental information processing included two MAPK signaling pathway genes. They played a crucial role in xenobiotic metabolic processes (Figure 3E). A total of 12/13 KEGG pathways were metabolism processes. The other representative significantly changed genes were nine glutathione S-transferases genes, two uridine diphosphate glycosyl transferase genes, three alcohol dehydrogenase genes, two hematopoietic prostaglandin D synthase genes, two sorbitol dehydrogenase genes, and one ecdysteroid UDP glucosyl transferase gene.

A total of 466 shared, upregulated DEGs between the three different time points with the control samples were analyzed (Figure 4A). These upregulated DEGs were categorized into 58 GO terms (25 biological processes, 12 molecular functions, and 21 cellular components) by GO classification. Among them, the top three GO classifications were metabolic process (202 DEGs, GO:0008152), catalytic activity (190 DEGs, GO:0003824), and cellular process (188 DEGs, GO:0009987) (Figure 4B). A total of 256 DEGs were categorized into 546 GO terms by GO enrichment analysis (*p* < 0.05). Carboxylic acid metabolic process (72 DEGs, GO:0019752) and oxidation-reduction process (60 DEGs, GO:0055114) were prominent GO terms according to GO enrichment analysis (Figure 4C). Furthermore, 101 DEGs were categorized into 10 KEGG pathways (*p* < 0.05, *q* < 0.05). Among them, nine glutathione S-transferases genes (GST) and two hematopoietic prostaglandin D synthase genes (HPGDS) were significantly changed, which contributed to drug metabolism by cytochrome P450 (24 DEGs, ko00982), metabolism of xenobiotics by cytochrome P450 (24 DEGs, ko00980), and glutathione metabolism (17 DEGs, ko00480) determined by KEGG enrichment analysis (Figure 4D,E).

With the passage of exposure time, the number of DEGs of the PWNs also changed. The functional descriptions of the DEGs are shown in Appendix A. A total of 70 DEGs were exclusively upregulated when the PWNs were exposed to *A. robusta* for 24 h (Figure 4A). These DEGs mainly included the Aldo-keto reductase gene (AKR), aldehyde dehydrogenase gene (ALDH), dopa decarboxylase gene (DDC), glutathione S-transferases gene (GST), and sphingosine 1-phosphate receptor gene (SLPR), which belong to the pathway of neuron apoptotic process and development. These genes were concentrated on tryptophan metabolism (ko00380), glycerolipid metabolism (ko00561), glycolysis/gluconeogenesis (ko00010), longevity-regulating pathway (ko04212), and axon regeneration (ko04361) according to the KEGG analysis (Appendix A) and the regulation of neuronal apoptotic processes (GO:0043523), neuronal apoptotic processes (GO:0051402), motile cilium (GO:0031514), and CatSper complex (GO:0036128) according to the GO analysis (Appendix A). In summary, the movement and immunity of the PWNs were affected when they were exposed to *A. robusta* for 24 h.

A total of 81 DEGs were exclusively upregulated at 30 h (Figure 4A). These DEGs mainly included the glutathione peroxidase gene (GPX), glutathione S-transferases gene (GST), phosphatase gene (PHO), cytochromep-450 reductase gene (CPR), and viral cathepsin gene (VCATH), which belong to the cellular matrix and metabolism pathway. These genes were concentrated on arachidonic acid metabolism (ko00590), lysosome (ko04142), glutathione metabolism (ko00480), and riboflavin metabolism (ko00740) according to the KEGG analysis (Appendix A) and structural constituent of cuticle (GO:0042302) and extracellular matrix (GO:0031012) according to the GO analysis (Appendix A). In short, xenobiotic metabolism-related pathways, especially the fatty acid metabolism and energy metabolism of the PWNs, were affected when the PWNs were exposed to *A. robusta* for 30 h.

In the last stage, a total of 83 DEGs were exclusively upregulated at 36 h. These DEGs mainly included actin genes (ACT), viral cathepsin genes (VCATH), alkaline protease receptor genes (APR), acyl sphingosine amidohydrolase genes (ASP), glycerol-3-phosphate acyltransferase gene (GPAT), phospholipase D genes (PLD), and lipid phosphate phosphohydrolase genes (LPIN), which belong to the organism catabolic process and lysosome pathway. These genes were concentrated on phagosome (ko04145), autophagy (ko04140), lysosome (ko04142), glycerophospholipid metabolism (ko00564), phototransduction (ko04745), and the hippo-signaling pathway (ko04391) according to the KEGG analysis (Appendix A) and fibrillar collagen trimer (GO:0005583) and dense body (GO:0097433) according to the GO analysis (Appendix A). In brief, the epidermis of the PWNs was degraded when they were exposed to *A. robusta* for 36 h.

### 3.3. Gene Set Enrichment Analysis

GSEA is a feasible way to determine whether genes within each gene set are enriched in the upper or lower part of a phenotypic-correlation-ranked gene list. Moreover, it is also an effective means for determining the effect of synergistic changes in genes within this gene set on phenotypic changes. Thus, in this study, we performed GSEA based on the magnitude of changes in gene expression according to the results of RNA sequencing. The dataset has 16,252 genes for which gene symbol collapsing was requested. The sequencing results were further analyzed for KEGG enrichment by GSEA. The remaining 6859 gene sets were used in the GO analysis, and 111 gene sets were used in the KEGG analysis after the gene sets were size-filtered (min = 15, max = 500). In CK vs. T24 (*B. xylophilus* exposed to *A. robusta* for 24 h), 8159 (50.2%) genes were tagged in CK, with a correlation area of 32.1%, and 8093 (49.8%) genes were tagged in T24, with a correlation area of 67.9%. In CK vs. T30 (*B. xylophilus* exposed to *A. robusta* for 30 h), 7888 (48.5%) genes were tagged in CK, with a correlation area of 32.2%, and 8364 (51.5%) genes were tagged in T30, with a correlation area of 67.8%. In CK vs. T36 (*B. xylophilus* exposed to *A. robusta* for 36 h), 9273 (57.1%) genes were tagged in CK, with a correlation area of 42.4%, and 6979 (42.9%) genes were tagged in T36, with correlation area of 57.6%. In T24 vs. T30, 7825 (48.1%) genes were tagged in T24, with a correlation area of 49.6% and, 8427 (51.9%) genes were tagged in T30, with a correlation area of 50.4%. In T24 vs. T36, 10,204 (62.8%) genes were tagged in T24, with a correlation area of 66.2%, and 6048 (37.2%) genes were tagged in T36, with a correlation area of 33.8%. In T30 vs. T36, 10,404 (64.0%) genes were tagged in T30, with a correlation area of 63.8%, and 5848 (36.0%) genes were tagged in T36, with a correlation area of 36.2%.

By ranking the normalized enrichment scores (NES) from highest to lowest (NOM *p* < 0.05; FDR *q* < 0.05), several significantly enriched signaling pathways were identified (Figure 5). GSEA for KEGG and DEGs identified a vast number of functional categories that appear to be potentially affected in the PWNs exposed to *A. robusta* for 24 h: arachidonic acid metabolism (ko00590, *p* = 0, *q* = 0), glutathione metabolism (ko00480, *p* = 0, *q* = 3.76 × 10^−4^), drug metabolism by cytochrome P450 (ko00982, *p* = 0, *q* = 2.50 × 10^−3^), metabolism of xenobiotics by cytochrome P450 (ko00980, *p* = 0, *q* = 2.86 × 10^−3^), tyrosine metabolism (ko00350, *p* = 1.12 × 10^−3^, *q* = 4.18 × 10^−3^), fatty acid elongation (ko00062, *p* = 0, *q* = 1.12 × 10^−2^), fatty acid metabolism (ko01212, *p* = 0, *q* = 1.18 × 10^−2^), butanoate metabolism (ko00650, *p* = 4.61 × 10^−3^, *q* = 1.80 × 10^−2^), and inositol phosphate metabolism (ko00562, *p* = 0, *q* = 2.55 × 10^−2^) (Figure 5A,D–F). In the PWNs exposed to *A. robusta* for 30 h, a number of functional categories appeared to be potentially affected: arachidonic acid metabolism (ko00590, *p* = 0, *q* = 0), metabolism of xenobiotics by cytochrome P450 (ko00980, *p* = 0, *q* = 3.56 × 10^−4^), drug metabolism by cytochrome P450 (ko00982, *p* = 0, *q* = 4.40 × 10^−4^), glutathione metabolism (ko00480, *p* = 0, *q* = 9.72 × 10^−4^), and tyrosine metabolism (ko00350, *p* = 1.13 × 10^−3^, *q* = 1.65 × 10^−3^) (Figure 5B,D–F). In the PWNs exposed to *A. robusta* for 36 h, a number of functional categories appeared to be potentially affected: arachidonic acid metabolism (ko00590, *p* = 0, *q* = 0), glutathione metabolism (ko00480, *p* = 0, *q* = 0), metabolism of xenobiotics by cytochrome P450 (ko00980, *p* = 0, *q* = 0), drug metabolism by cytochrome P450 (ko00982, *p* = 0, *q* = 1.97 × 10^−4^), tyrosine metabolism (ko00350, *p* = 1.30 × 10^−3^, *q* = 1.20 × 10^−3^), lysosome (ko04142, *p* = 0, *q* = 1.16 × 10^−2^), fatty acid metabolism (ko01212, *p* = 0, *q* = 3.84 × 10^−2^), and fatty acid elongation (ko00062, *p* = 0, *q* = 4.19 × 10^−2^) (Figure 5C–F). Next, we focused our analysis primarily on the genes involved in processes known to have a significant impact on nematode physiology during fungal infection.

### 3.4. Weighted Gene Co-Expression Network Analysis

After removing null and outlier genes, 15,067 genes were selected for WGCNA. The power value chosen for this analysis was 7 (Figure 6A). When the average gene connectivity was 7, the value decreased to zero indefinitely (Figure 6B). A total of 32 modules labelled with different colors were delineated. These modules contained between 67 and 3167 genes (Figure 6C). After the initial module’s partitioning, we obtained the initial module’s results. Then, we merged the modules with similar expression patterns based on the similarity of each module’s feature values to 0.7 to obtain the final partitioned module (Figure 6D).

In total, 12 samples were used to construct a map of the correlations between the gene expression patterns and the trapping processes. The relationships between each module and the trapping process were evaluated based on significant correlations (Figure 7A). The results showed that three modules had highly specific correlations (|r| > 0.60 and *p* < 0.001), including mediumpurple2, thistle1, and darkslateblue. Red represents the most positive module, and darkslateblue is correlated with trapping (r = 0.81, *p* = 0.005).

The genetic composition of the specific modules most closely associated with PWN exposed to *A. robusta* was investigated by calculating the ME values of the individual modules and corresponding genes to investigate the network-specific properties of gene significance (GS) and module membership (MM). Meanwhile, the MM values and K.in values were highly correlated, indicating that the gene was more significant for the trait than the given module. The higher the level of GS and MM of a gene, the more influential its characteristics. A higher r-value and a smaller *p* value for a given module indicate that the module members are more representative of that module’s characteristics. Compared to other modules, darkslateblue (cor = 0.59, *p* = 1 × 10^−200^) played more important biological roles in processes associated with the PWNs’ exposure to *A. robusta* (Figure 7B). Trends in the genetic change of darkslateblue showed that CK and T36 were higher than T24 and T30 (Figure 7C).

### 3.5. Analysis of Darkslateblue Module

To better understand the biological processes of the darkslateblue module, we parsed the module by GO and KEGG analysis. Most of the genes were enriched in terms of the amino acid metabolic process (GO:0006575) and small molecule metabolic and biosynthetic processes (GO:0044281, GO:0044283) GO terms (Figure 8A,B). In the KEGG analysis, genes were also enriched in terms of amino acid metabolism, including arachidonic acid metabolism (ko00590) and lysosomes (ko04142) of cellular processes. However, we also found activities related to xenobiotic metabolism and redox, including peroxisome and drug metabolism by cytochrome P450 (ko00982) and arachidonic acid metabolism (ko00590) in the biological process category (Figure 8C,D).

These genes were flavin-containing monooxygenases genes (FMO), glutathione S-transferase genes (GST), uridine diphosphate glycosyltransferase genes (UGT), alcohol dehydrogenase genes (ALDH), epoxide hydrolase genes (BXYJ_LOCUS5661), hydroxyphenylpyruvate dioxygenase genes (BXYJ_LOCUS5779), carbonyl reductase genes (BXYJ_LOCUS6352), and thrombin AT genes (BXYJ_LOCUS7889) from 15 metabolism-related genes in the darkslateblue module according to the results from Cytoscape3.9.1. These 15 hub genes were associated with 69 genes from 148 pairs of relationships. Meanwhile, the results of the network showed that the genes of localization, biological regulation, and signaling had the most complex interrelationship with other categories (Figure 9, Appendix A).

### 3.6. Validation of Transcriptome Data by RT-qPCR

Generally, the results of the gene expression-level validation via RT-qPCR of specific functional DEGs in PWNs at three different times showed the same conditions as the RNA sequencing data. Alcohol dehydrogenase (ALDH) was significantly upregulated when exposed to *A. robusta* (*p* < 0.001), for which the difference multipliers at the three time points were 163.84 (T24), 85.67 (T30), and 203.85 (T36); cathepsin L-like cysteine proteinase (BXYJ_LOCUS12540) showed significant upregulation when exposed to *A. robusta* (*p* < 0.001), for which the difference multipliers at the three time points were 29.02 (T24), 23.74 (T30), and 29.66 (T36); and fatty-acid and retinol-binding protein (BXYJ_LOCUS15290) showed significant upregulation when exposed to *A. robusta* (*p* < 0.001), for which the difference multipliers at the three time points were 11.80 (T24), 8.51 (T30), and 13.30 (T36). Moreover, five glutathione S-transferases (GST) exhibited significant upregulation during exposure to *A. robusta*. This suggested that PWNs respond to fungal stress primarily through metabolism. The trends in the relative expression levels of the DEGs were consistent with the RNA-seq results. This indicated that the RNA-seq data obtained in this study were accurate, reliable, and had referential significance (Figure 10).

## 4. Discussion

*Bursaphelenchus xylophilus* represents a critical threat to global forestry and ecosystems, and biocontrol methods are becoming an increasingly attractive means to fight PWN infestation [44]. The development of nematophagous fungi as biological control agents and their infection processes have been researched extensively [45]. In this study, we described the interactions between PWNs and the nematophagous fungus *A. robusta*. Since its discovery, *Arthrobotrys robusta* has been widely used to control various nematodes, such as *Haemonchus contortus* [46], *Haemonchus placei* [47], *Strongyloides papillosus* [48], *Strongyloides stercoralis* [49], and *Strongyloides venezuelensis* [50]. *Arthrobotrys* sp. seem to produce very few traps constitutively but form bountiful ones in the presence of nematodes [10]. Herein, we examined the morphological changes in PWNs when exposed to *A. robusta* at three time points via morphological observation. Before the PWNs were exposed to *A. robusta*, no traps were observed. When the nematodes were in close proximity to the fungus, traps formed gradually until the PWNs were eventually caught. The morphological characteristics of the PWNs exposed to *A. robusta* were similar to those of *C. elegans* exposed to *Drechmeria coniospora* [51], and nematode exposure significantly increased trap formation.

Several reports have shown that nematophagous fungi release a series of metabolites in the presence of nematodes [52]. These metabolites enter the nematodes and alter the magnitude of related gene expression, thereby affecting their survival at the molecular level. Other reports have shown that the process whereby nematophagous fungi form abundant traps to capture and consume nematodes is not only a physical process but is also accompanied by a series of chemical reactions [53,54]. Polyketide–terpenoid hybrids [55,56] and 6-methylsalicylic acid [57], which were produced by *Arthrobotrys oligospora,* displayed moderate nematode-inhibitory ability. Cyclohexanamine, cyclohexanone, and cyclohexanol, which are produced by *Duddingtonia flagrans*, inhibited the egg-hatching process of *Meloidogyne incognita* [58]. Some metabolites produced by fungi stimulate the nervous system of nematodes upon entry into their body [59]. However, little is known regarding gene expression in nematodes exposed to nematophagous fungi. Even less is known about the gene expression of PWNs exposed to nematophagous fungi.

We have supplemented this component of the gene expression of nematode exposure to nematophagous fungi by transcriptomic analyses of PWNs’ exposure to *A. robusta*. In this study, the vast majority of DEGs (>65%) were upregulated. The number of DEGs in the PWNs exposed to *A. robusta* was significantly higher than that in the PWNs exposed to nematicides. This study was conducted to investigate the molecular mechanisms of *A. robusta* predating on PWNs. Many DEGs of PWNs undergo differential expression when these nematodes are in the presence of the fungus in question. Through the results of the GO analysis generated via DEGS, we found that the genes of the structural constituent of the cuticle (GO:0042302) and collagen trimers (30 DEGs, GO:0005581) were significantly changed. This result indicated that the epidermal structure of PWNs was significantly damaged when the PWNs were exposed to *A. robusta*. Many genes of metabolic processes in the PWNs were significantly changed according to the GO and KEGG analyses. Among them, the xenobiotic metabolism pathway contained a number of DEGs that were divided into three phases [60]: cytochrome P450 genes (CYP450) and flavin-containing monooxygenase genes (FMO) in the modification phase; glutathione S-transferase genes (GST), uridine diphosphate glycosyl transferase genes (UGT), ecdysteroid UDP glucosyl transferase genes (EGT), alcohol dehydrogenase genes (ADH), and sorbitol dehydrogenase genes (SODH) in the conjugation phase; and ABC transporter genes B (ABCB) in the excretion phase. The xenobiotic metabolism pathway has been proven to be prominent in PWNs in response to nematicide stress [61,62]. This study is the first to show that the pathway was also prominent in PWNs in response to nematophagous fungi.

Nuclear hormone receptors (NHRs), as important nervous receptors, mediate the expression of xenobiotic-metabolizing-enzyme-related genes (XMEs) when *C. elegans* is exposed to *Penicillium brevicompactum* [63]. To date, XMEs have been more frequently studied with respect to nematodes’ response to nematicide stress [60,62]. Xenobiotics are excreted from the bodies of nematodes throughout the three phases of xenobiotic metabolism. Xenobiotics were modified in phase I metabolism reactions by cytochrome P450 [64] and flavin-containing monooxygenases [65]. Phase II metabolic enzymes conjugate the metabolic intermediates of phase I metabolism reactions into more soluble forms by glutathione S-transferases (GST), uridine diphosphate glycosyltransferase (UGT), alcohol dehydrogenase genes (ALDH), short-chain dehydrogenase (SDR), and alcohol dehydrogenase (ADH) [66,67]. Subsequently, xenobiotics are excreted from nematodes’ bodies by ATP-binding cassette (ABC) [68]. Nematodes maintain normal activities when xenobiotics are excreted from their bodies, and the concentration of xenobiotics is reduced. Moreover, the expression of lysosomes facilitates the breakdown of proteins into amino acids [69,70]. This has a positive regulatory effect on autophagy. Therefore, combined gene expression and phenotypic changes in PWNs when exposed to *A. robusta* at three time points were recorded. As a result, we found that morphological changes in the PWNs and alterations in the expression of genes were closely related. At 24 h, the PWNs had just entered the traps of the nematophagous fungi and the neural and immune genes of the PWNs were upregulated. Donnell et al. demonstrated that microbiology can promote the fitness of both the host and the microorganism by overriding the host’s control of a sensory decision [71]. Otherwise, nematodes can be involved in innate host defense via the nervous and immune systems to enhance the duration of infection [72,73,74]. This phase shows that nematophagous fungi may stimulate the nervous and immune systems of PWNs when the interaction of nematophagous fungi with nematodes has just occurred.

At 30 h, the traps of the nematophagous fungi contracted, and the bodies of the PWNs were significantly constricted. At the same time, the genes related to xenobiotic metabolism were significantly upregulated. As mentioned in the literature review, a large number of studies have reported that nematodes can excrete xenobiotic substances via the xenobiotic metabolism pathway. Therefore, we ensured that the nematophagous fungi released certain substances into the bodies of the nematodes while contracting their traps, which activated the xenobiotic metabolism pathway of the nematodes in this phase. These results reflect those of Wallace et al., who also found that fungi produce toxins that cause mitochondrial stress and are toxic to nematodes, and that these toxins are cleared by the induction of XMEs in the intestine [63].

The bodies of the PWNs degraded when they were exposed to *A. robusta* at 36 h, and lysosomal and autophagy-related genes of PWNs were upregulated. Previous studies indicated the occurrence of the lysosome-mediated lipolysis of *Meloidogyne incognita* [75]. A strong relationship between lysosomes and autophagy has been reported in the literature [76]. Meanwhile, the upregulation of autophagy-related genes increases proteasomal degradation [77]. Therefore, the results of this study indicated that nematophagous fungi regulate the autophagy–lysosomal degradation pathway of PWNs in this phase.

Interestingly, there are six genes with high correlation coefficients. BXYJ_LOCUS6666 (basic leucine zipper transcription factor) is a bZIP transcription factor that has been reported to act as both an activator or repressor in the network [78]. BXYJ_LOCUS6703 (ankyrin) is usually used as an anchor molecule that assists the specific positioning of various membrane proteins in cells [79]. BXYJ_LOCUS7900 (P-type ATPases) generates and maintains chemical gradients across cellular membranes by translocating cations, heavy metals, and lipids [80]. BXYJ_LOCUS8025 (innexin) is involved in the formation of gap junctions [81]. BXYJ_LOCUS9127 (major facilitator) facilitates the transport across cytoplasmic or internal membranes for a variety of substrates [82]. BXYJ_LOCUS9801 (cytochrome P450) catalyzes a variety of oxidative reactions of a large number of structurally different endogenous and exogenous compounds in organisms from all major domains of life [83]. These six highly connected genes are the bridges that signal the different stages of PWNs when exposed to *A. robusta*. In future research, we will explore their functions and obtain their specific roles in PWNs when exposed to *A. robusta*.

## 5. Conclusions

In summary, a working model of PWNs when exposed to *A. robusta* was proposed. The mobility of the PWNs increased when they had just entered the traps of *A. robusta*. Furthermore, *A. robusta* produced metabolites that stimulate the nervous system of this nematode and trigger its immune response. The PWNs recognized xenobiotics when they were exposed to *A. robusta* for 30 h. The PWNs produced energy via lipid metabolism and energy metabolism-activated xenobiotic metabolism, which excreted xenobiotics from the nematodes’ bodies. Lysosomal and autophagy-related genes were upregulated, and the nematodes’ bodies disintegrated when they were exposed to *A. robusta* for 36 h. This research will facilitate the identification of genes associated with the resistance of nematodes and provide a broad basis for understanding the molecular and evolutionary mechanisms of nematode–nematophagous microbe interactions. Additionally, this research will provide crucial information for the biological control of plant-parasitic nematodes.

## Figures and Tables

**Figure 1 cells-12-00543-f001:**
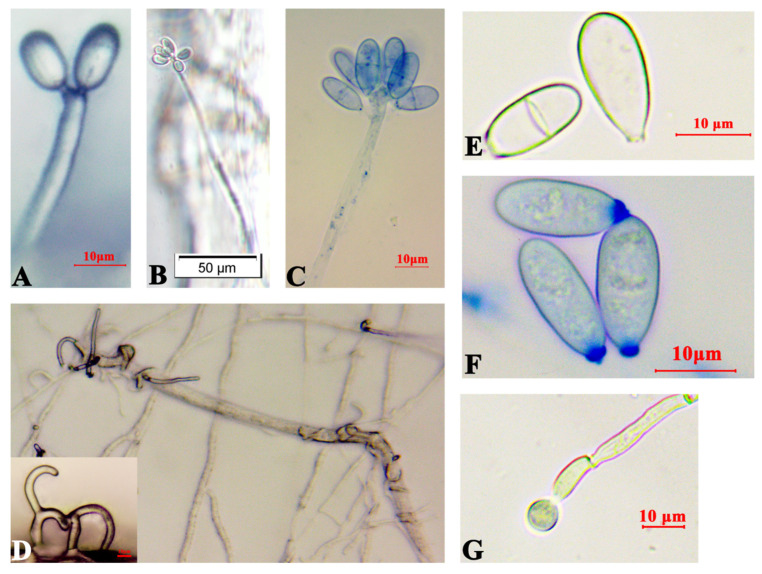
Morphology of *Arthrobotrys robusta* on LCMA. (**A**–**C**) Mycelium and conidiophores; (**D**) Bundled PWNs and three-dimensional fungal cords; (**E**,**F**) Conidia; and (**G**) Chlamydospores.

**Figure 2 cells-12-00543-f002:**
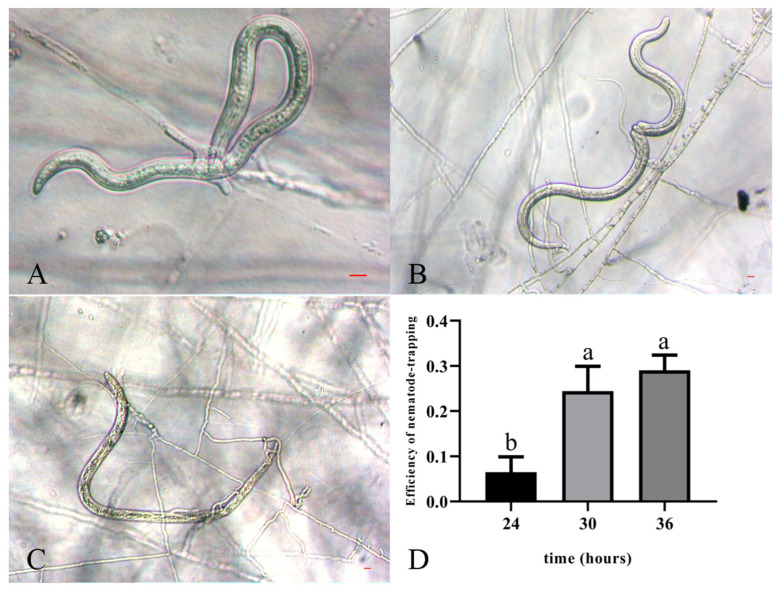
Trapping process of PWNs exposed to *A. robusta*. (**A**) The formation of the trap when PWNs were exposed to *A. robusta* for 24 h. (**B**) The trap’s contraction when PWNs were exposed to *A. robusta* for 30 h. (**C**) The increase in the size of the trap when PWNs were exposed to *A. robusta* for 36 h. Bar = 10 μm. (**D**) Nematode-trapping efficiency of PWNs exposed to *A. robusta*.

**Figure 3 cells-12-00543-f003:**
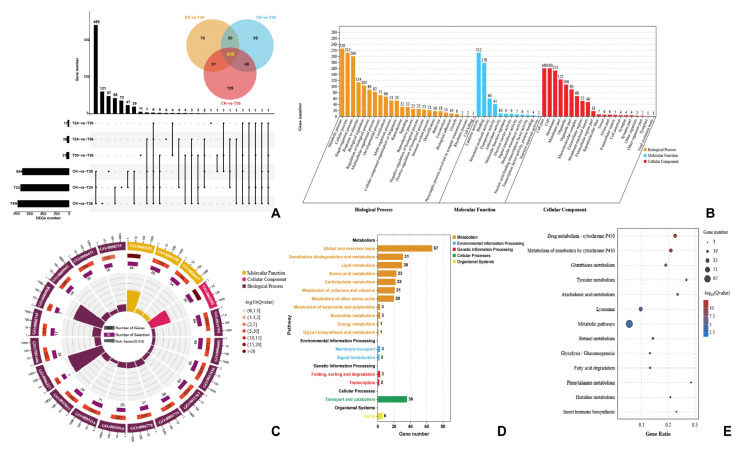
DEGs of PWNs when exposed to *Arthrobotrys robusta*. The upset plot and Venn diagram show the number of DEGs in each group (**A**). Set Size represents the number of DEGs. GO classification (**B**) and enrichment analysis (**C**) of DEGs (*p* ≤ 0.05). The first circle: different colors represent different ontologies; the second circle: the number of genes in the GO term; the third circle: purple represents the proportion of DEGs; and the fourth circle: the number of differential genes in the GO term divided by the number of background genes in the GO term. KEGG pathway classification (**D**) and enrichment analysis (**E**) of the DEGs.

**Figure 4 cells-12-00543-f004:**
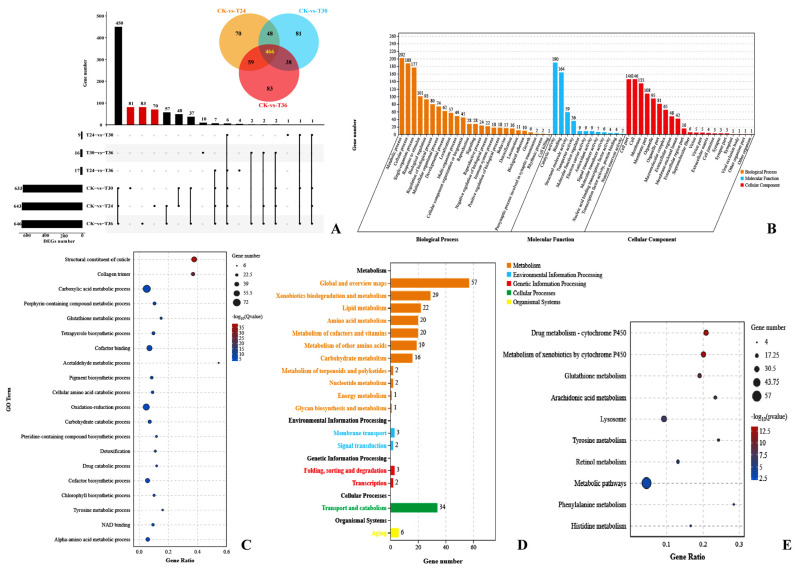
Upregulated DEG expression patterns of PWNs when exposed to *Arthrobotrys robusta*. The upset plot and Venn diagram show the number of upregulated DEGs in each group (**A**). Set Size represents the number of DEGs. GO classification (**B**) and enrichment analysis (**C**) of DEGs (*p* ≤ 0.05). KEGG pathway classification (**D**) and enrichment analysis (**E**) of the DEGs.

**Figure 5 cells-12-00543-f005:**
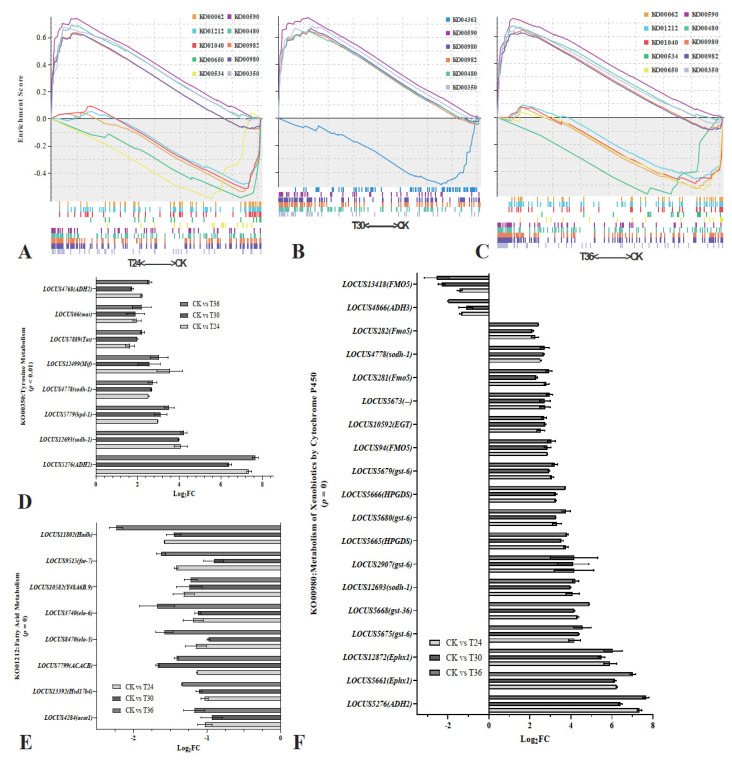
Gene set enrichment analysis (GSEA) of DEGs by KEGG. (**A**) GSEA of CK vs. T24 by KEGG; (**B**) GSEA of CK vs. T30 by KEGG; and (**C**) GSEA of CK vs. T36 by KEGG. NOM *p* < 0.05; FDR *q* < 0.05. (**D**) KO00350 pathway gene expression levels in RNA-seq; (**E**) KO01212 pathway gene expression levels in RNA-seq; and (**F**) KO00980 pathway gene expression levels in RNA-seq.

**Figure 6 cells-12-00543-f006:**
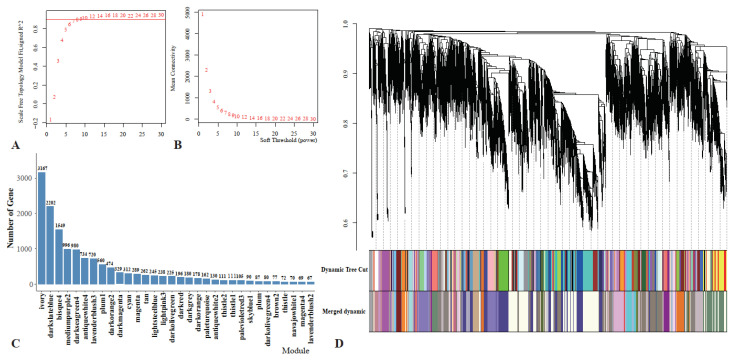
Co-expression network analysis diagram. (**A**) The scale-free fit index versus soft-thresholding power; (**B**) changes in average gene connectivity at different powers; (**C**) the number of genes in each module; and (**D**) clustering dendrograms of genes and different colors represent different modules.

**Figure 7 cells-12-00543-f007:**
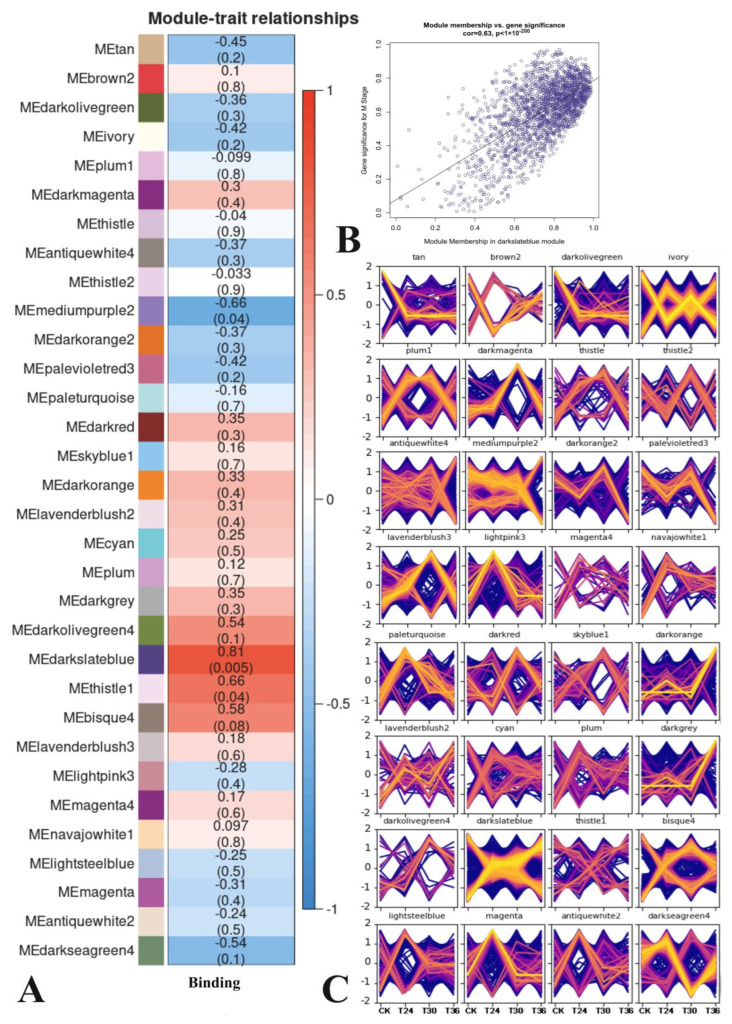
Correlation diagrams of RNA-seq. (**A**) Module trait relation. (**B**) Scatterplot of MM and GS correlations for the most positively correlated module. The dots indicate genes, and the diagonal lines indicate the fitted lines of correlation. (**C**) Genes expression pattern in each module. The yellow lines represent patterns of gene expression that match the pattern most closely, whereas red denotes similar patterns, and blue denotes less similar patterns.

**Figure 8 cells-12-00543-f008:**
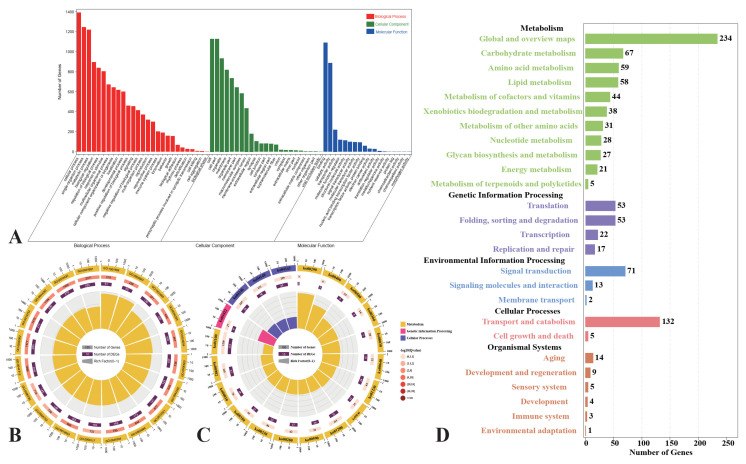
Genes expression patterns of darkslateblue. (**A**) Go classification; (**B**) GO enrichment analysis (*p* ≤ 0.05). The first circle: different colors represent different ontologies; the second circle: the number of genes in the GO term; the third circle: orange represents the proportion of up-regulated genes, while purple represents the proportion of down-regulated genes; and the fourth circle: the number of differential genes in the GO term divided by the number of background genes in the GO term. (**C**) KEGG enrichment analysis (*p* ≤ 0.05). The first circle: different colors represent different ontologies; the second circle: the number of genes in the KEGG pathway; the third circle: orange represents the proportion of up-regulated genes, while purple represents the proportion of down-regulated genes; and the fourth circle: the number of differential genes in the KEGG pathway divided by the number of background genes in the KEGG pathway. (**D**) KEGG pathway classification.

**Figure 9 cells-12-00543-f009:**
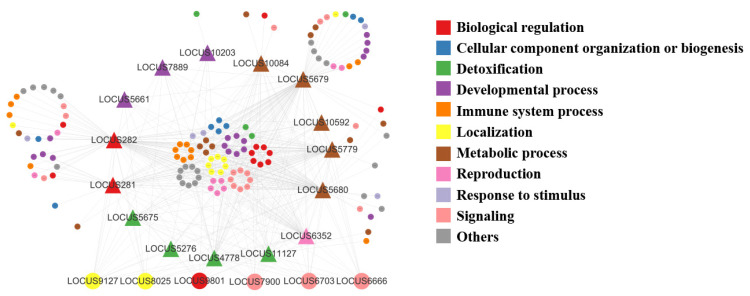
Gene co-expression pattern network developed using Cytoscape3.9.1. Different colors represent different pathways, and the larger circles represent transcription factors.

**Figure 10 cells-12-00543-f010:**
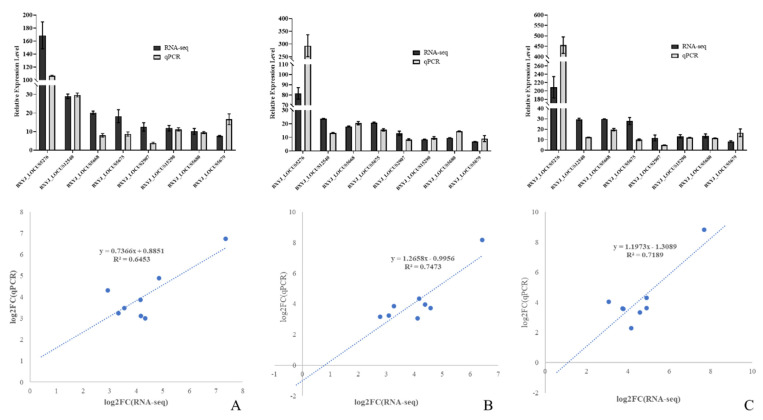
Expression of differentially expressed genes in CK vs. T24 (**A**), CK vs. T30 (**B**), and CK vs. T36 (**C**).

## Data Availability

Not applicable.

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
