# Peer review of "Molecular Defense Response of Bursaphelenchus xylophilus to the Nematophagous Fungus Arthrobotrys robusta"

_cells, 2023, doi:10.3390/cells12040543_

Round 1
Reviewer 1 Report
The submitted manuscript which entitled ' Molecular Defense Response of Bursaphelenchus xylophilus to the Nematophagous Fungi' is bring an important research topic. However, i very much encourage the authors to bring it to a more mature levels via include more details in the introduction as well as in the materials and methods so makes it easy for applied researchers to understand the topics. The english language and the scientific writing must be improved. I will be happy to read this manuscript once those major concerns are handled.
Author Response
Thanks for your recognition and support of our work. We added more details in the introduction and methods. We hope those make the easier for readers to understand this study. Otherwise, we would like to apologize for the poor English academic writing. We have invited our native English-speaking colleagues to improve the English writing for the manuscript. See our latest revised manuscript for more details.

Reviewer 2 Report
Review of “Molecular Defense Response of Bursaphelenchus xylophilus to the Nematophagous Fungi”
Original and relevant topic to the field, exploring new avenues of biological control of the pinewood nematode with the nematophagous fungus Arthrobotrys robusta. Authors shed light on important pathogenic mechanisms of the fungus in the presence of the nematode, which could help develop new sustainable management strategies in the future.
Major comments:
Title: rather misleading, as the authors studied the molecular response of Bursaphelenchus xylophilus to a specific antagonistic fungus, Arthrobotrys robusta, and not nematophagous fungi in general. That should be cleared out to avoid confusion, so I propose the title “Molecular Defense Response of Bursaphelenchus xylophilus to the Nematophagous Fungus Arthrobotrys robusta”.
Abstract: the English needs some polishing.
Introduction: well-structured and references are appropriate.
Material & Methods: experimental design is good, experiments are reproducible and methods are appropriate. Proper references to previously published methodology. Sound and comprehensive statistical analysis.
Results & Discussion: results, including tables and figures, are clearly labelled and nicely presented. Logical interpretation of results and conclusions.
Specific comments:
Abstract – page 1
Lines 16-18: past tense in the first two sentences is inappropriate, so it needs to be changed to something like: “Bursaphelenchus xylophilus causes pine wilt disease which poses a serious threat to forestry ecology around the world.”
Lines 17-18: I would rephrase the sentence to “Microorganisms are environmentally friendly alternatives to chemical nematicides in controlling B. xylophilus in a sustainable way.”
Line 18: “In this study, we found nematophagous fungi Arthrobotrys robusta from the xylem of diseased [...]” – only one species was isolated, so “fungi” needs to be replaced by “fungus”, and “found” preferably by “isolated”.
Lines 19-20: “Then we observed B. xylophilus exposed to A. robusta from active to dies each 6 h.” The sentence is quite confusing. Were the nematodes exposed to active compounds produced by the fungus, or in direct contact with the mycelium? I’m guessing it’s the latter, so I would rephrase to “We observed/recorded nematophagous activity of A. robusta on the PWN after just 6 h.” Then again, I’m not sure if this is what authors meant, but it’s what I can make out from the sentence.
Line 22: that A. robusta produced in the meantime.
Line 22: After 30 hours – hours is written out here, when it’s abbreviated above, so I would use 30 h.
Line 23: and we were able to identify xenobiotics
Lines 23-24: when starting the sentence, write out the name of the species rather than the abbreviated form. Rephrase to “Bursaphelenchus xylophilus activated xenobiotic metabolism, which expelled the xenobiotics from its body, by providing energy through lipid metabolism and energy metabolism.” – what do authors mean by energy metabolism?
Line 31: Rephrase to “This study proposes a model that mobility improved whenever B. xylophilus entered the traps of A. robusta.”
Lines 32-33: “The model will make it easier to find genes linked to resistance of nematode [...]” – genes of resistance of the nematode to the fungus? This part is confusing. I would simplify the sentence: “The model will provide a solid foundation to understand the molecular and evolutionary mechanisms underlying interactions between nematodes and nematophagous fungi.”
Introduction
Line 43: Have caused and still do! I would rephrase to “Plant-parasitic nematodes (PPNs) cause immeasurable losses to agriculture and forestry worldwide.”
Line 47: cause pine wilt disease [...]
Lines 50-52: I don’t think the idea that synthetic nematicides are powerful tools in controlling nematodes was acquired as a general truth in recent years... This has been established for a long time, so the sentence needs to be rephrased: Synthetic nematicides are powerful tools to manage PWD and are widely used in many Asian countries.
Line 54: “[...] for pollution-free methods to control PWNs.” – for sustainable methods to control PWNs
Lines 55-56: a class of fungi that use nematodes as nutritional sources, have gained attention
Lines 56-57: “Arthrobotrys, as a most representative genus of nematophagous fungi” – based on what? What makes it the most representative genus of nematophagous fungi? I would simplify the sentence: Among nematophagous fungi, the genus Arthrobotrys has received extensive attention from researchers. Need some references here to back this up.
Line 59: A. cladodes (abbreviated form)
Line 60: whether the vitro temperature changes or not – the vitro? In vitro?
Lines 62-63: For accurate and effective control of PPNs [...]
Lines 74-75: [...] PWNs were templated to explore the molecular regulatory mechanisms of nematode response to A. robusta by transcriptomic analysis
Lines 76-77: [...] PWNs were exposed to the nematophagous fungus
Lines 77-78: [...] it was specifically demonstrated how the fungus affects the virulence mechanisms of PWNs
Material and Methods
Line 82: seeing as only one was isolated, it’s fungus and not fungi. This needs to be changed consistently (where appropriate) throughout the manuscript.
Lines 84-85: Botrytis cinerea fungal mat rather than moss. nematodes were reared in vitro (Petri dishes) or in vivo? This part if confusing: “[...] washed them with 84 M9 buffer from the plants.” “The nematophagous fungi were identified by molecules and morphology [...]” – see comment for line 82. I would rephrase the entire sentence: The nematophagous fungus was characterized molecularly and morphologically, and identified as A. robusta.
Line 92: agarose gel
Results
Line 175: P. massoniana (abbreviated form)
Line 176: Mycelium is white and sparse
Line 182: septate spores is repeated twice
Lines 183-190: I would start each of these sentences by clearly stating which time point these results belong to, otherwise it is counterproductive and readers will have to skip directly to the graph in order to understand what authors mean. For instance: At 24 h, PWNs were found inside A. robusta’s traps and the nematode-trapping efficiency was 6.47% (fig. 2A). At 30 h post inoculation, trap structures contracted and fixed nematodes. After 36 h, the movement of PWNs was restricted by several trap structures and the nematode-trapping efficiency was 24.42% (fig. 2B), with a significant nematode color deepening (what do authors mean by this? Color change?), and the nematode-trapping efficiency was 29.03% (fig. 2C and 2D).
Figure 1: “Morphology of A. robusta on LCMA” rather than “Morphological of A. robusta on LCMA.”
Figure 1D: the scale is missing.
Figure 2: “Trapping process of PWNs when exposed to A. robusta” rather than “Trapping process of PWNs were exposed to A. robusta”.
Figure 2D: how was the efficiency of nematode-trapping determined? There is no reference to that in the methods. How many replicates per time point were considered? How many nematodes were inoculated onto the A. robusta mycelium to determine this? Are the differences shown in the graph statistically significant?
Discussion
Line 427-428: Bursaphelenchus xylophilus represents a critical peril to global forestry and ecosystems, and biocontrol methods are increasingly attractive as a means to fight PWNs infestation.
Line 430: are researched deeply/extensively
Line 431: Arthrobotrys needs to be written out at the beginning of the sentence
Line 434: I would remove the sentence “Nematophagous fungi capture nematodes by producing traps.”.
Lines 435-436: Simplify: Arthrobotrys sp. seem to produce very few traps constitutively, but form bountiful ones in the presence of nematodes.
Lines 437-438: Here, we examined the morphological changes of PWNs when exposed to A. robusta, at three time points, by morphological observation.
Lines 438-439: Before PWNs were exposed to A. robusta, no traps were observed
Lines 439-440: The sentence is confusing, so I rephrased it: When the nematodes were in the close vicinity of the fungus, traps formed gradually until PWNs eventually got caught.
Line 440: what morphological characteristic?
Line 442: probiotic effect on trap formation? I would rather write “potentiated trap formation”.
Line 445: while producing a large number of feeding structures – what kind of feeding structures? I would simply write “Several reports have shown that nematophagous fungi release a series of metabolites in the presence of nematodes” (this already implies the production of trapping structures or other mechanisms that they use to antagonize nematodes and ultimately feed on them).
Lines 445-447: Some sentences are repeated and I simplified it: These metabolites enter the nematodes and alter the magnitude of related gene expression, thereby affecting their survival at the molecular level.
Lines 454-455: Some metabolites produced by fungi stimulate the nervous system of nematodes upon entry into their body
Lines 461-462: It is a dangerous affirmation, especially considering it is not backed by any previous study to compare this claim to. I would remove “Therefore, we think that the threat from nematophagous fungi is greater than the threat from nematicides.”
Lines 463-464: The sentences were too repetitive, so I simplified them: This study was conducted to investigate the molecular mechanisms of A. robusta predating on PWNs. Many DEGs of PWNs undergo differential expression when the nematodes are in the presence of the fungus.
Author Response
Thanks for your recognition and help in our work. We have explained the change made, including the exact location where the change can be found in the revised manuscript.

Reviewer 3 Report
The paper could be interesting for controlling of Bursaphelenchus xylophilus by using nematophagus fungi, but it is not well presented . The abstract and the introduction were not clear and easy to understand. Thus I suggest the authors to revise the English grammar with a native speaker in order to clearly present the problematic and then the results. I can follow the experiments because I am involved in such kind of reasearch but in my opinion in this paper is very difficult to understand the results obtained at morphological and molecular level. I suggest also torevise the discussion section because it seems to me a list of genes. The manuscript can be interesting above all because the trap fungi can be used as control of Bursaphelenchus but it is difficult to understand.
Author Response
I apologize for the confusion we caused. First, we have rewritten the abstract, introduction, and discussion. Meanwhile, we have revised the results section to make it clearer. We have revised the English grammar with native English-speaking colleagues.
I hope our changes will improve the quality of our manuscripts and help the reviewer to understand our research. We hope to explore the changes in the nematode by combining the observation of their movement under the fungus with the time series of RNA-seq data.

Round 2
Reviewer 1 Report
The manuscript entitled as 'Molecular Defense Response of Bursaphelenchus xylophilus to the Nematophagous Fungi' has been significantly improved.
minor correction - latin names in the figures captions should be written in full.
Author Response
Thank you for reviewing our manuscript and for the constructive comments, which greatly helped us to improve the manuscript. We have heavily revised our experiments. We have revised the manuscript in accordance with the reviewer’s comments.
